# Nurturing the Early Life Gut Microbiome and Immune Maturation for Long Term Health

**DOI:** 10.3390/microorganisms9102110

**Published:** 2021-10-07

**Authors:** Shaillay Kumar Dogra, Chung Cheong Kwong, Dantong Wang, Olga Sakwinska, Sara Colombo Mottaz, Norbert Sprenger

**Affiliations:** Nestlé Institute of Health Sciences, Société des Produits Nestlé S.A., 1000 Lausanne 26, Switzerland; ShaillayKumar.Dogra@rd.nestle.com (S.K.D.); William.KwongChung@rd.nestle.com (K.C.C.); Dantong.Wang@rdls.nestle.com (D.W.); Olga.Sakwinska@rdls.nestle.com (O.S.); Sara.ColomboMottaz@rd.nestle.com (S.C.M.)

**Keywords:** microbiome, nutrition, immunity, infants, toddlers

## Abstract

Early life is characterized by developmental milestones such as holding up the head, turning over, sitting up and walking that are typically achieved sequentially in specific time windows. Similarly, the early gut microbiome maturation can be characterized by specific temporal microorganism acquisition, colonization and selection with differential functional features over time. This orchestrated microbial sequence occurs from birth during the first years of age before the microbiome reaches an adult-like composition and function between 3 and 5 years of age. Increasingly, these different steps of microbiome development are recognized as crucial windows of opportunity for long term health, primarily linked to appropriate immune and metabolic development. For instance, microbiome disruptors such as preterm and Cesarean-section birth, malnutrition and antibiotic use are associated with increased risk to negatively affect long-term immune and metabolic health. Different age discriminant microbiome taxa and functionalities are used to describe age-appropriate microbiome development, and advanced modelling techniques enable an understanding and visualization of an optimal microbiome maturation trajectory. Specific microbiome features can be related to later health conditions, however, whether such features have a causal relationship is the topic of intense research. Early life nutrition is an important microbiome modulator, and ‘Mother Nature’ provides the model with breast milk as the sole source of nutrition for the early postnatal period, while dietary choices during the prenatal and weaning period are to a large extent guided by tradition and culture. Increasing evidence suggests prenatal maternal diet and infant and child nutrition impact the infant microbiome trajectory and immune competence development. The lack of a universal feeding reference for such phases represents a knowledge gap, but also a great opportunity to provide adequate nutritional guidance to maintain an age-appropriate microbiome for long term health. Here, we provide a narrative review and perspective on our current understanding of age-appropriate microbiome maturation, its relation to long term health and how nutrition shapes and influences this relationship.

## 1. Co-Development in Our Gut as a Strategy for Life

The first 1000 days, roughly from conception to the child’s second birthday, represent a series of critical windows of opportunities for healthy development and long-term health. During this period important foundations for immediate and long-term health are built. Human biology is driven by interaction and communication processes including those from and to our abiotic and biotic environment. Among the latter, the diversity and abundance of different microbes are a key element. A distinctive microbial community present in a defined habitat and their activity is generally referred to as microbiome. It is composed of the microbiota, an assembly of microorganisms such as bacteria, archaea, protozoa, fungi and algae along with the products of their activity, such as microbial components, metabolites and mobile genetic elements (e.g., transposons, phages) all embedded in a specific habitat [1]. Most importantly, the microbial activities within a community, their interactions among them and with their host habitat drive the dynamics and establishment of a stable ecology. While the critical importance of the gut microbiome for health has been recognized for at least a century with the groundbreaking work of Metchnikoff [2], only recent years saw an increasing appreciation of the critical window represented by the highly dynamic maturation of the infant gut microbiome. Lessons to this end stem primarily from associations seen in case–control observational studies and from clinical observations that involve early life microbiome disruptors (e.g., antibiotics) and modulators (e.g., nutrition). The extent to which microbiome changes have a causal link to the observed developmental and health outcomes requires mechanistic insight in the clinical setting or substantiation using preclinical models such as gnotobiotic animals (transfer of infant microbiota) to replicate clinical phenotypes.

Epidemiologic surveys led Barker and Osmond to indicate that poor nutrition in early life increases the risk for later cardiovascular health and possibly other health conditions such as bronchitis, for example [3]. Insight from nutritional studies with preterm born infants led Alan Lucas to formulate the concept of ‘programming’, suggesting that a stimulus or insult during a critical period of development exerts not only immediate, but likely also lasting long-term effects that contribute to health and disease [4,5]. Together with experimental findings from rodent models, this concept was further developed and is today largely recognized and known as DOHAD (developmental origins of health and disease) [6].

As recently reviewed, accumulating evidence demonstrates that the gut microbiome contributes to this early life imprinting, particularly through its effects on the developing immune system [7,8]. The establishment of the microbial-host symbiosis relies on the mutualistic co-development of the host, microbiota and immune system from its very first encounter and sets the stage towards a healthy trajectory. This ultimately begs the question of what are the processes that are in place early in life to establish a successful partnership? To answer this question, the two following concepts are currently discussed in the field [7,9]: (i) There are critical time windows for microbial colonization and concomitant immune development. In these windows, non-redundant priming of the immune system by the microbiome directs the behavior of the immune system later in life. (ii) The right symbionts need to colonize the intestine at the right time. Such an orchestrated sequence of microbial colonization may ensure a cascade of immune development in a timely manner.

Figure 1 provides a general overview on microbiome-relevant determinants and features at different stages during development in relation to immune development and nutrition.

## 2. Establishment of the Gut-Microbiome Mutualism

The exposure to microbiome elements begins prenatally through maternal microbial components and metabolites. However, if and to what extent the fetus is exposed to life microbes, challenging the paradigm of a sterile womb, is highly debated [10,11]. Exposure of the normally developing fetus to metabolites and components, possibly including DNA yet only sparsely found, from the maternal microbiome is plausible, while exposure to life microbes is not. During gestation, microbial metabolites affect the developing fetus as suggested by different studies. For example, intake of a poor and inadequate diet characterized by high sugar and low fiber content, likely affecting the gut microbiome, was associated with a higher risk for severe viral respiratory infections in infants [12]. It is not established whether these effects were due to an in utero priming through microbial metabolites or to an inadequate seeding of microbes from mother to infant. In another study focusing on preeclampsia condition, the microbially produced maternal short chain fatty acid (SCFA) acetate was linked to fetal plasma acetate and regulatory T-cell development with lasting postnatal effects [13]. In concordance with this study, gestational carriage of *Prevotella copri*, an acetate producing microbe, is strongly linked to a decreased risk of infant food allergy manifestation independent of infant *Prevotella copri* carriage [14]. Of note, different clades of *Prevotella copri* are described and their presence in populations with a Westernized lifestyle seems strongly reduced [15]. Possibly, such *Prevotella copri* clade diversity informs as to why in some studies *Prevotella copri* has been related to inflammatory conditions while in other studies it was linked with health promoting effects [16]. These observations may partly be due to different microbial component exposure in utero and to dysbiotic microbiota seeding at birth. As recently shown in mice, higher maternal SCFA (e.g., acetate) was shown to reduce the allergy risk of their offspring mediated through regulatory T-cell expansion and function [17]. Similarly, in a rodent model, maternal intake of aryl hydrocarbon receptor (AHR) ligands reduces postnatal TLR4 signaling in pups and thereby protects from NEC [18]. Microbial metabolites such as indole-lactate derived from tryptophan can activate AHR, indicating that maternal microbial metabolites may also influence neonatal gut immune compartments through AHR signaling. At birth, the microbiome-host interaction intensifies and becomes more direct with the initiation of a *bona fide* microbial colonization of the infant [19,20,21]. At delivery, the microbiologically essentially sterile infant is exposed to a multitude of microbes from the mother and the environment. In the context of microbiome inheritance, it is important to note that significant maternal gut microbiota changes were reported to occur over pregnancy, such as an expansion of some taxa, such as Actinobacteria and Proteobacteria, as well as a higher interindividual diversity [22] towards the third trimester, in line with the physiological changes and adaptions in immune function. Interestingly, women’s diet during pregnancy relates to some extent to the observed microbial communities in their infant when vaginally delivered, highlighting the importance of vertical microbiota transmission [23]. Primarily, the infant gut is colonized by maternal gut and skin microbes, during birth and breastfeeding, with additional microbial input from the environment including siblings, father and other household members [19,24,25,26].

‘Birth seeds, breast milk feeds’ captures the current thinking on the early life microbiome development in the infant gut. This concept, with microbiome inheritance and nurturing as foundations, can be further characterized by progression and maintenance [27,28,29]. The gut is colonized progressively and sequentially with distinct microbial populations during infancy and early childhood from an aerobic to an anaerobic milk-oriented microbiome first, then to a more diverse adult-like microbiome [24]. This concept is reinforced by the observation that the intra-individual alpha diversity of an infant’s gut microbiome is low during early life and increases over time with a concomitant reduction in inter-individual beta-diversity [24]. Not surprisingly, the compositional changes also reflect to some extent microbial functional competencies, as illustrated by marked changes in the abundance of microbial carbohydrate active enzymes (CAZymes) [30] and other metabolic pathways [29]. These illustrate microbiome changes as an adaptation to exogenous factors such as diet as well as endogenous to the microbial ecology including the interaction with the developing gut. Gut immune components, such as secretory immunoglobulin (Ig) A and defensins, together with epithelial and mucous glycosylation patterns change while the gut develops and likely play an important role in setting the stage for the development of host-microbiome mutualism.

The intestinal barrier built of mucus and underlying epithelial cells is primarily considered a physical barrier contributing together with numerous immune defense components, including secretory IgAs and defensins, to modulate the microbiome and host relationship. The physicochemical properties of the mucous barrier are largely driven by the abundance of diverse glycan structures [31]. Developmental changes in glycosylation patterns on mucosal surfaces are expected to guide microbiome maturation including mucous-associated microbes that potentially interact closest with the developing gut and immune system [32]. Specific microbes dwell on glycans exposed on mucous, which can be considered as feeding from inside, as opposed to feeding through diet, and this mechanism may greatly contribute to establishing an appropriate gut ecology.

## 3. Microbiome Maturation Windows of Opportunity for Immune Development

Appreciation of the importance of the microbiome in immune system development stems mainly from observations related to microbiome risk factors or perturbations. These can be exogenous factors, such as inadequate microbial component exposure during gestation, Cesarean (C)-section birth and antibiotic exposure, or endogenous conditions, such as gut developmental immaturity due to premature birth [33,34,35,36,37].

Preterm infants generally show a delayed and irregular microbiome maturation compared to term infants [33,34]. For term infants, C-section delivery, often in conjunction with antibiotic use, is one of the primary disruptors of the early life microbiome maturation [35]. Of note, an elective and emergency C-section may not bear the same risks for an appropriate microbiome establishment [36] with an emergency C-section possibly having additional underlying risks, besides the birth mode per se. Often, a delay in establishing a ‘Bifidobacterium peak’ around 3 months of age is observed in infants exposed to microbiome disruptors, such as preterm birth, C-section birth and early antibiotic use [38]. While the exact role of the *Bifidobacterium* peak is not fully established, a promising hypothesis is that *Bifidobacterium*-specific metabolites contribute to the immune and mucosal barrier function development. For example, the tryptophan metabolite indole-lactate produced by *Bifidobacterium* species, such as *B. longum*, *B. bifidum* and *B. breve* typically present during the breastfeeding period, was shown to activate the aryl hydrocarbon receptor (AHR) and hydrocarboxylic acid receptor 3, thereby contributing to appropriate mucosal immune development [39,40]. *Bifidobacterium longum* subsp. *infantis* related metabolites including indolelactate were also reported to skew T cell polarization, providing a molecular link to immune regulation [41]. In addition, acetate, a typical *Bifidobacterium* metabolite, has been shown to protect from infections in basic research models, in part through the type one interferon signaling pathway [42,43].

Importantly, such risk factors during early life have been correlated in a compelling number of studies and meta-analyses to elevated risk for inappropriate immune competence and/or metabolic development [44,45,46,47,48,49]. Accordingly, C-section birth was considered a genuine, although modest, risk for allergic outcomes in children, with asthma having an odds ratio (OR) of about 1.16 (95% CI 1.14–1.29), food allergy/food atopy an OR of 1.32 (95% CI 1.12–1.55) and allergic rhinitis an OR of 1.23 (95% CI 1.12–1.35), while for eczema and atopic eczema as well as inhalant atopy, no significant risk was seen in the meta-analysis [48]. Similarly, a recent population-based study reported a link between the incidence of asthma at 5 years of age and antibiotic use during the first year, which altered microbiome structures [49]. This confirms earlier observations on microbiota disruption driven by frequent antibiotic treatment during neonatal life leading to immune dysregulation and increased susceptibility to allergy later in life [50]. Such observations highlight the importance of the early life microbiome for appropriate immune competence development.

Following the exclusive breast milk feeding period in early life, we increasingly appreciate the weaning period as being critically important in the imprinting of the immune system and representing one of the windows of opportunity. As mentioned, the cessation of breastfeeding and consequent transition to other food types leads to increased bacterial diversity and functional maturation and expansion of the gut microbiota [24]. Appropriate microbiome diversification and progression is critical for appropriate immune competence development as suggested by the observed associations to atopy and asthma later in life [27,51]. Pre-clinical evidence in mice demonstrates that increased bacterial richness during the weaning period leads to a strong immune reaction characterized by a transient pro-inflammatory IFNγ/TNFα-driven immune response accompanied by the induction of microbiota-driven RORγt+ Foxp3+ regulatory T cells (Treg) [52]. Interfering with this so-called ‘weaning reaction’ results in an inappropriate imprinting of the immune system and subsequent increased susceptibility to allergy, colitis and cancer later in life. Furthermore, microbial colonization after the weaning period cannot compensate for the lack of microbiota-induced immune stimulus in early life and the weaning reaction.

Since microbiome–host immune system interactions in early life dictate long-term immune functionality [53], it is a highly conceivable postulate that the right symbionts need to colonize the intestine at the right time. Recently, an epidemiologic study described higher gut microbiota maturity below 10 weeks of age and lower gut microbiota maturity above 30 weeks of age as risks for atopic dermatitis [27]. Whether the ‘right’ symbiont or commensals can generally be described through their taxonomy or functional capacity needs to be established. A timely choreography of microbial colonization ensures that microbiota-derived signals do not overwhelm the developing immune system in early life and that these signals can be interpreted correctly for the development of both the innate and adaptive immune system. In parallel, the intestinal T cell compartment in neonates is characterized by high levels of suppressive regulatory T cells (as opposed to adults) to control immune responses and maintain gut immune homeostasis [54]. Ultimately, what we define as an immature immune system during neonatal life [55] may in fact be the result of the orchestrated co-development of the microbiota and immune system with both key components being in synchrony.

## 4. Specific Immune Triggers and Homeostasis for the Gut Microbiome Development

Given the co-development of the microbiome and the immune system, it is paramount that microbial-derived signals are interpreted correctly by gut epithelial sensor cells and the immune system. To this purpose, innate immune cells use a diversity of pattern recognition receptors (PRRs) to recognize conserved molecular patterns of microbes and these receptors are also expressed in a timely manner [56,57]. PPRs include Toll-like receptors (TLR), nucleotide-binding oligomerization domain (NOD)-like receptors and leptin-like receptors. Signaling of these receptors may then act as developmental triggers for the gut microbiome.

In mice, TLR4 expression in the intestinal epithelium is downregulated before birth coupled with an upregulation of TLR9 expression (which counteracts LPS-mediated TLR4 signaling) to prevent inappropriate immune stimulation [58,59]. Consequently, there is a lack of intestinal epithelial cell response to LPS stimulation that is thought to promote host-microbial immune homeostasis after birth, a period during which the first microbes start to colonize the intestine [60]. Similarly, human blood monocytes are reprogrammed to promote immune tolerance following exposure to TLR4 ligands S100A8/A9 (calprotectin) [61]. Consistent with the need for a timed expression of different PRRs for appropriate microbiota-immune signaling, TLR5 expression in murine gut epithelium is restricted to the neonatal period [56]. Intestinal epithelial TLR5 is critical in shaping the microbiome composition in early life. A lack of TLR5 expression in neonatal mice results in an altered microbiota that persists during adulthood, resulting in increased risk of disease development [62,63,64].

The compartmentalization of PRRs adds another layer of regulation to ensure correct interpretation of microbial-derived signals. In humans, epithelial expression of TLR1-10 can be found in the small intestine, whereas the colonic epithelial cells display preferential expression of TLR3 and, to a certain extent, TLR2 and TLR4 [65,66]. This reduced TLR4 expression coupled with lower levels of MD2, an element needed to react to LPS, leads to intestinal epithelial cells hypo-responsiveness to LPS stimulation, which is known to induce a pro-inflammatory response [67]. Additionally, location of epithelial TLR4-MD2 complexes in the intestine are restricted to the crypts, resulting in strategic compartmentalization that may promote immune silence to gut microbial-derived signals and restrict immune activation solely to organisms capable of reaching the deeper layers of the mucosal epithelium [68].

Additionally, different specialized sensor cells are interspersed in the gut epithelia. Among them are enteroendocrine and Tuft cells. These sensor cells have multiple functions, such as local regulatory roles for gut function, systemic regulation of metabolism and food intake, for example. Recently, thanks to single cell gene expression analysis, Tuft cells regained interest as they specifically express several receptors and pathways relevant for immune regulation [69,70,71,72]. For instance, the specific Tuft cell succinate receptor SUCNR1 (GPR91) through which the microbially produced succinate triggers a type two immune response also leads to the expansion of mucus producing Paneth cells [73]. Interestingly, the highest level of succinate in stool is reported in the first 6 months of age [74]. The role of Tuft cells as immune sentinels in early life to shape the developing immune system remains to be studied in detail. An important role may be expected in view of the microbiome communities that change rather dramatically over the first postnatal year.

## 5. The Microbiome and Immune System Symbiosis and Its Long-Term Implications 

Given the impact of the host microbiome in guiding immune system functionality, it is critical for the host to ‘learn’ how to tolerate the microbiome. While it is abundantly evident that early events of the microbiota-immune system co-development sets the stage for immune imprinting, maintaining this symbiosis provides the foundation to sustain health throughout life.

The epithelial barrier, with its immune sentinels, provides and maintains a critical interface separating the microbiome from the underlying immune cells. Gut microbes induce IL-22 production by intestinal innate lymphoid cells resulting in fucosyltransferase Fut2 mediated fucosylation of epithelial cells. These fucosylated carbohydrate moieties can, in turn, be used by the gut microbiota with fucose catabolizing capabilities as an energy source. This mechanism was shown to promote host-microbiome symbiosis during *Citrobacter rodentium* infection [75] and enhance protection following *Salmonella typhimurium* exposure [76]. In infants, a possible link between intestinal fucosylation deficits and necrotizing enterocolitis (NEC) was recently shown and replicated in an animal model [77]. In preterm infants, generally a delay in microbiome maturation is observed [33,34]. However, the extent to which the preterm microbiome may relate to intestinal fucosylation and risk for NEC remains unknown.

Adaptive immune mechanisms involving B cell antibody production is also important in establishing and maintaining a host–microbiota–immune dialog. The first source of antibodies in early life is derived from the mother’s milk. This passive transfer of maternal IgA in cooperation with IgG was shown to dampen mucosal T cell responses against newly acquired microbes in mice [78]. Consistent with this, murine T cells still display a naïve phenotype up to the third week of life, despite the increased exposure to microbial antigens [79]. In infants, naïve T cells are phenotypically distinct to adult naïve T cells with no sign of converging to an adult phenotype up to 6 months of life [80]. Within the same cohort, it was further shown that the presence of HMO utilizing bifidobacteria correlated with a decrease in systemic inflammation as well as intestinal T helper (Th) 2 and Th17 responses [41].

With the introduction of solid food in the diet, there is a dramatic expansion in the diversity of gut microbes and antigens [81]. IgA is the most predominant antibody expressed in the human intestine and plays a key role in shaping the gut microbiome composition [82]. Using antibody repertoire sequencing, it was found that long-lived memory B cells diversify their repertoire to adapt to the presence of microbiota [83]. By shaping the microbiome composition and promoting microbial diversity, IgA indirectly promotes the generation of Foxp3+ Treg cells, which are key for immune regulation and the host’s ability to tolerate the presence of microbes [84]. These regulatory T cells, in turn, regulate IgA responses by differentiating into T helper follicular cells to perpetuate a positive feedback loop, which all together promote gut–immune homeostasis. Of note, immune encounters with bacterial-derived antigens favor the generation of Treg in the intestine [85]. The unique repertoire of intestinal Treg as opposed to their counterparts in other organs demonstrates that the microbiota directly shapes the repertoire of these cells and reinforces the concept that the immune system adapts to tolerate the presence of non-self-antigens [85]. On the other hand, higher levels of the microbial produced lipid 12,13-diHOME reduced the expansion of intestinal Treg and was related to higher risk for later asthma manifestation [86,87].

Learning to tolerate the microbiome provides long-term health benefits by keeping a balance between immune silence to innocuous stimuli and immune responsiveness to pathogens. Although the mechanisms of microbiota-induced protective effects are not completely elucidated, it is generally accepted that microbially derived metabolites play a key role in this process. As these metabolites can reach systemic compartments, the induced immune benefits can extend beyond the gut. Several microbial produced metabolites, such as desaminotyrosine (DAT), promote the induction of type I interferon responses, which protect mice against influenza infection and decrease infection-mediated lung immunopathology [88]. Importantly, treatment with DAT prior but not after influenza infection prevented mortality in mice, suggesting that such microbiota metabolites promote a protective immune response. Similarly, the pretreatment of mice with a high fiber diet that leads to increased microbial short chain fatty production or providing short chain fatty acids prior to infection improves viral clearance following influenza or respiratory syncytial virus infection by enhancing a local antiviral response while dampening immunopathology [42,89]. In addition, propionate administration to neonatal mice with depleted plasmacytoid DCs ameliorates pneumovirus-induced bronchiolitis and subsequent asthma [90]. In short, keeping a balance between immune stimulation and silence is key to be ready to respond when needed. This principle is nicely exemplified by the gut–lung axis driven protection from viral respiratory infection via type 1 IFN protective stimulation through gut microbial metabolites [42].

Overall, these data add to the concept that an established mutualism with the microbiome primes the host immune system to be ready for a pathogen encounter and appropriate immune competence development. The extent to which fighting or feeding the microbiome by endogenous as well as exogenous factors directs the establishment of a mutually beneficial relationship warrants to be understood as an important basis to design meaningful preventive and corrective interventions.

## 6. Modelling the Gut Microbiome Maturation toward a Universal Discriminant of Health 

As mentioned before, the early life microbiome changes rapidly and dramatically in the first years of life, reflecting to some extent the infant’s development. Hence, the quest is to characterize an appropriate gut microbiome maturation profile that starts with a rather limited number of taxa during the first months after birth and then expands to reach a richer and eventually more stable microbiome community after about 3 to 5 years [27,29,91,92,93,94]. Different metrics can be used to describe and characterize this dynamic process.

The microbiome development can be described by trajectories of individual taxa or their functional capacity [94]. This works well when focusing on higher taxonomic levels or when focusing on selected taxa and features only. However, with more detailed information such as species, subspecies or even strain level information, this approach becomes limited. Additionally, microbes dwell in close relations within diverse communities interdependent with the infant host. Eventually, this needs to be considered when describing the gut microbiome maturation, for example, by describing covarying taxa over time that allow distinct microbial networks to be described [95].

Diversity assessments considering richness, evenness, as well as phylogenetic distance measures between taxa also help to understand and describe the development of the microbiome as a whole [29]. Similar, models such as Dirichlet multinomial mixtures (DMM) can identify patterns or microbiome communities and describe their change and succession over time [29,96]. However, DMM is strongly data structure dependent. For example, the number of time bins can change the optimal number of distinct communities, which limits its use to describe more universally valid and dataset independent trajectories.

Alternatively, machine learning classification methods can help to further build our understanding of important features for specific time bins and health conditions during development, therefore allowing us to describe the normal gut microbiome maturation trajectory. One such approach that gained much interest in microbiome research is based on Random Forest (RF) models. Analogous to the WHO child growth curves used as a calibration measure to monitor a child’s anthropometric development over time, the microbiome data collected longitudinally were used to predict a child’s age [97]. The derived ‘microbiota age’ or ‘microbiota maturity’ is based on an abundance of key age discriminant microbes or functional capacity modules. Plotted against the infant’s chronological age, it proved useful to track normal microbiota development over time and to differentiate the mode of delivery, breastfeeding status, specific health conditions such as atopic dermatitis and asthma as well as malnutrition status [27,29,97,98,99,100]. Malnutrition, for example, makes infants fall off the normal early life microbiome trajectory. Interestingly, the microbiota age-based trajectory also helped to monitor how a nutritional intervention in malnourished infants corrects their microbiome toward the normal trajectory [97].

In a pioneering study of Bangladeshi infants, about 20 bacteria taxa were chosen as important by a machine learning algorithm to capture the relation of microbiota with infant age from birth to 24 months [97]. Among the top ones were *Faecalibacterium (F.) prausnitzii*, *Ruminococcus* species, *Bifidobacterium longum*, *Lactobacillus* species and other *Bifidobacterium* species. Using the same approach in another study of different geographic locations, a similar number of taxa were considered important by the machine learning algorithm to capture the relation of microbiota with infant age from 3 to 40 months [29]. Among the top taxa were *Anaerostipes* species, *Ruminococcus* species, *Faecalibacterium* species, *Eubacterium hallii* group, Lachnospiraceae, *Subdoligranulum* species, *Alistipes* species and *Bifidobacterium* species. Of note, only some of the key microbiota taxa picked by the model were the same between these studies. On the technical side, the models captured about 70% of the variance using a similar number of 20 odd bacteria as features to predict the microbiota age.

As indicated, the taxa richness increases with a marked shift seen after 6 months of age [29,94,97]. Identified key age discriminant taxa higher in the first 6 months and decreasing thereafter are *Bifidobacterium* species, such as *B. longum*, but also *Staphylococcus*, *Enterobacter* and *Escherichia* [29,97]. The other main age discriminant taxa increase from 10 months with *Faecalibacterium, Anaerostipes* and Lachnospiraceae coming up earlier and *Eubacterium*, *Ruminococcus* and *Alistipes* later. 

The discrepancy of taxa used in the gut microbiota modeling between different studies is perhaps due to environmental differences, although methodological variations also need to be considered [101,102]. Differences among geographies have been reported for early life microbiota but tend to diminish with age [103]. Yet, also in adults some variation pertains as their gut microbiome associates with geographic location [104,105]. To overcome possible geographic differences, in a meta-analysis, Ho et al. used only the common bacterial taxa across seven different studies to describe the microbiota age using an RF model [99]. Among the top taxa were *Blautia* species, Lachnospiraceae, *Prevotella* species, Clostridiales, *Staphylococcus* species, *Dialister* species, *Lactobacillus* species, *Haemophilus* species, *Bifidobacterium* species, *Actinomyces* species, *Dorea* species and others. This modified model still explained 65% of the variance for the data from Bangladeshi infants for which the original model explained 70% of the variance.

These models are useful to describe the microbiome maturation, yet they are not perfect as they, in general, capture only about 70% of the variance. The curves are sigmoidal with a linear range of prediction from about 6 months to 18 months followed by a plateau from about 2 years onwards [29,97]. Hence, a good fit between microbiota age and chronological age is achieved primarily during the linear part of the trajectory curve of infancy and toddlerhood. Similarly, when the microbiota age is translated into microbiota-for-age z-scores (MAZ), the first 2 years of age showed for most infants a MAZ with a standard deviation within −2 to +2. Thereafter, the standard deviation went into large negative values. One possible explanation is that the approach of modeling the microbiota age as a proxy for infant gut microbiota development only works reliably in an age span characterized by significant compositional changes with time. Indeed, this may reflect that the microbiota will become more stable after about 2 years of age [29]. In adults, due to a more stabilized microbiota, RF-based microbiota age modeling was far less accurate in predicting the chronological age [106].

Describing microbiota maturation that is universally valid and accurate enough to differentiate the impact of infant maturity at birth, mode of delivery, diet and current or future health conditions is a general ambition. To overcome the possible issue with taxonomic differences across different geographies, a model may be built based on universally present taxa [99] or, alternatively, on functional data, such as metagenomics functionality [100] and possibly metabolites. Assuming the functionalities are redundant among different taxa and better reflect the physiology of the gut ecology, such an approach may be more accurate and universal. Gut microbiota functional capacity data from Bangladeshi infants were recently used to model microbiota age [100]. Some of the top pathway modules used by this model were lipoate biosynthesis, pyridoxine and pyridoxal uptake, folate biosynthesis, riboflavin biosynthesis, folate uptake and biotin uptake. The model accuracy was comparable to the microbiota age derived from taxonomic data. It remains to be seen whether models based on microbiome functional data from different geographies have these same pathway modules and, thus, present more similarity with each other as compared to models based on microbiota compositional data.

As mentioned, mode of delivery, breastfeeding status, nutritional status and health conditions such as asthma and atopic dermatitis were related to an altered gut microbiota maturation trajectory. Several statistical methods may be considered to assess the impact of different factors and conditions on the early life microbiome maturation trajectory [107,108,109]. It remains to be established if describing the microbiome maturation through the ‘microbiota age’ robustly captures alterations related to different health conditions as was shown for breastfeeding status using a meta-analysis approach [99]. To this end, its prognostic power would be of highest interest in order to design corrective actions. For certain health conditions with a supposed contribution of the gut microbiome development, we suggest advanced analytical models need to be envisaged that integrate most differentiating microbiome features in the modeling.

Timely progression is characteristic of the microbiome maturation, and it is conceivable that passing too fast, too slow or not at all through a specific microbiome ecology influences subsequent dynamics. The first indications to this end were observed using microbiota age in relation to atopic dermatitis [27]. Lastly, interdependence and interactions between different microbes and the infant host affect the microbiome dynamics and ecology in the gut. Network analysis of covarying microbiome features is one approach to capture microbial dynamics [95]. Integration of infant host parameters in such models may be one way forward to further establish the mechanistic underpinning of the microbiome-host interactions for appropriate development and health. Lastly, nutrition and dietary patterns need to be integrated as major modulators of the gut microbiome maturation.

## 7. Infant Feeding Recommendations and Realities across the Globe

Adequate infant feeding is key for healthy growth and development. The WHO and UNICEF recommend an early initiation of breastfeeding within 1 hour of birth; exclusive breastfeeding for 6 months, continued breastfeeding until 2 years of age or beyond, with complementary diet introduced no later than 6 months of age [110]. While those recommendations have been adopted globally, about 60% of infants are still not breastfed in the first hour, and two in three infants are not exclusively breastfed for the recommended first 6 months [110].

At around 4 months of age, an infant’s renal and gastrointestinal physiology is mature to receive non-milk foods and at around 6 months of age, breast milk alone does not provide sufficient nutrients and energy for the infant’s needs [110,111]. Therefore, complementary food should be introduced to adequately support an infant’s growth and development. Complementary diets composed of food items other than breast milk or infant formula are introduced to provide nutrients and are often modified to a texture appropriate for the infant’s developmental readiness [111]. In Europe, the exact timing to initiate complementary feeding varies as some countries align with the WHO guideline to start complementary feeding around 6 months (e.g., UK and Sweden), while others recommend starting between 4 and 6 months of age (e.g., Belgium and Spain) in alignment with the European Society for Paediatric Gastroenterology, Hepatology and Nutrition (ESPGHAN) Committee on Nutrition. According to the ESPGHAN committee, complementary foods should not be introduced before 4 months and should not be delayed beyond 6 months of age, including allergenic foods [111]. Similar conclusions were reported in a recent review on foods related to IgE-mediated food allergies [112] and also in a scientific opinion published by the EFSA (European Food Safety Agency) Panel on Nutrition, Dietetic Products and Allergies (NDA) [113].

Recently, the EFSA NDA panel published a systematic review on the relationship between the timing of complementary food introduction and numerous aspects related to health conditions. These included physical development such as body weight and growth, risks of developing diseases or symptoms (e.g., allergies, coeliac disease and type 1 diabetes mellitus), nutrient status (e.g., iron), food preferences and eating behaviors later in life [113]. Based on the reviewed 283 publications, the panel concluded that there was no convincing evidence for adverse health effects following the introduction of complementary foods in term infants before 6 months of age, if nutrient composition and texture were appropriate, and foods were prepared using good hygiene practices. Additionally, the panel also found no clear evidence for any benefit of introducing complementary foods before 6 months of age, except for infants at risk of iron depletion. Accordingly, there is no uniform age to initiate complementary feeding. Rather, the appropriate age depends on each individual infant’s developmental stage, guided by the acquisition of the neuromotor skills necessary to progress from a liquid to a diversified, semi-solid and solid diet [114].

Other aspects of complementary feeding that need to be considered are feeding frequency, types of foods and their consistency. For the types of foods other than a mother’s milk, a minimum dietary diversity score of four is recommended in children 6–24 months of age. This means a child consumes four out of the following seven food groups: (1) grains, roots and tubers, (2) legumes and nuts, (3) dairy products (milk, yogurt, cheese), (4) meat, fish, poultry and liver/organ meats, (5) eggs, (6) vitamin A rich fruits and vegetables and (7) other fruits and vegetables [110]. A positive association between diet diversity score and mean micronutrient density adequacy was shown supporting the use of the diversity score as a diet quality indicator in populations not regularly consuming fortified foods [115]. As recommended by the China Nutrition Society, complementary feeding should start from iron rich foods, slowly increasing the diversity and allowing adaption to any new food item for around 2–3 days before introducing a new food item [116]. Similar, the Centers for Disease Control and Prevention (CDC) in the United States recommends introducing one food at a time allowing a 3–5-day interval before introducing a new one [117]. Concerning the energy provided by complementary foods, the China Nutrition Society recommends that for 7–12 months old infants, 30–50% of the energy should be provided through complementary foods [116]. Food and drinks with added sugars or that are high in salt should be avoided or limited before 12 months of age [117], and cow’s milk is among foods that should be limited in early age due to the increased risk of developing iron deficiency anemia [118].

Despite existing guidelines and recommendations for appropriate feeding habits, the reality is often different. In Mexico and the US, for example, already before their first birthday, 13 and 35% of infants consume undiluted fruit juice and sugar sweetened beverages, respectively, and many also consume cow’s milk [119,120].

Deviations from appropriate nutrition and recommendations may represent important risks for an age-appropriate microbiome development and maturation. This seems particularly relevant for recognized microbiome modulators such as dietary fibers. Their definition has evolved over the years and is subject to slightly nuanced definitions around the world. Yet, all contain the basic notion that dietary fibers are non-digestible carbohydrates and lignins, including all carbohydrate components occurring in foods that are non-digestible in the human small intestine and pass into the large intestine [121]. The recommended amount of fiber intake varies from country to country and ranges for adults from 18 to 38 g/day [121]. The reference intake in infants 6 to 12 months is 5 g/day [122]. Although the average dietary fiber intake in 6–12-month-old US infants increased over the years, from 5.3 g/day in 2002 to 6.5 g/day in 2016 [123], there are still over 25% of children below the recommended amount [124]. In China, the mean intake of fiber was reported as 2 g/day, much lower than the recommendation [125]. In 1–2-year-old toddlers, dietary fiber intake is about 10 and 5 g/day in the US and China, respectively [124,125]. However, due to the increased recommended adequate intake of 19g/day, over 80% of toddlers do not reach the recommendation in the US, China and Mexico [126,127]. Such dietary recommendations and especially the realities of infant and toddler diets regarding adequate intake need to be considered when investigating the early life microbiome development and maturation and their relation to health conditions.

## 8. Influence of Infant and Toddler Dietary Factors on the Gut Microbiome

### 8.1. Exclusive Breastfeeding Period and Early Infancy

Breast milk generally provides the reference nutrition for all infants, and exclusive breastfeeding for the first six months is recommended. The feeding mode, exclusive or partial breastfeeding versus no breastfeeding during early infancy, has well-recognized effects on the gut microbiome composition and function [99]. Hence, the health benefits observed for breastfed infants may partly be explained by a more appropriate gut microbiome maturation. Among those benefits are protection against infections, increased intelligence as well as probable reductions in becoming overweight and diabetes development later in life, while the situation for allergies such as asthma and atopies needs more clarification [128]. Breastfeeding, in particular when exclusive, leads to a higher abundance of bifidobacteria and lesser predominance of Enterobacteriaceae as well as lower alpha diversity compared to formula-fed infants [24,92,129,130].

The composition of consumed infant formula has also been shown in numerous observational cohort studies and randomized controlled trials to have a considerable impact on the microbiome. Among infants observed from birth to 8 months of age, a soy formula was associated with a diminished abundance of bifidobacteria and higher alpha diversity, in comparison to a cow’s milk-based formula [129]. A small study comparing older 12- to 24-month-old infants fed rice, soy and cow’s milk-based formulas showed a decreased abundance of *Bacteroides* and *Bifidobacterium* species associated with feeding a plant-based formula [131]. However, this observation may have been confounded by a suspected cow milk protein allergy, a primary reason to have used a plant-based formula. The addition of individual human milk oligosaccharides (HMOs) in starter infant formula resulted in a gut microbiome composition closer to that of breastfed infants with an increased abundance of *Bifidobacterium* species [132]. The HMO-driven microbiome community at 3 months of age was associated with a reduced risk of infections, as seen by the reduced need for antibiotics up to 12 months of age. This indicated that, indeed, a nutritionally modified early life microbiome maturation can impact later health. Other oligosaccharides such as bovine milk-derived galactooligosaccharides (GOS), for example, added to infant formula also showed a robust effect on the gut microbiome, increasing primarily bifidobacteria [133,134]. Most studies testing the effect of probiotics show more modest effects on the gut microbiome compared to prebiotics. However, studies in preterm and Cesarean section delivered infants found more pronounced effects [135,136]. Supplementation with a probiotic *Bifidobacterium longum* subsp. *infantis* in breastfed infants resulted in a higher abundance of the supplemented strain and an altered gut ecology [137].

### 8.2. From Introduction of Complementary Foods to Consumption of Family Foods 

Cessation of breastfeeding and introduction of complementary foods drive rather rapid and substantial changes in the infant gut microbiome [24,29]. Even though the introduction of complementary foods clearly remodels the gut microbiome, the knowledge on how particular foods and nutrients impact the developing microbiome is rudimentary and fragmented. Only a few longitudinal as well as cross sectional cohort studies examined associations between dietary intake and the gut microbiome with some evidence coming from intervention studies with prebiotics and probiotics. Intervention studies evaluating the impact of dietary intake are rare.

An interesting comparative study of 1–6-year-old children from Burkina Faso and Italy reported substantial differences in numerous taxa of the gut microbiota, as well as their metabolite profiles [138,139]. *Prevotella* was highly predominant, constituting more than 50% of the total microbiota in rural Burkina Faso, while it was nearly absent (<1%) in Italy. The opposite pattern was observed for Bacteroides. In parallel, short-chain fatty acids content was several folds higher in the stools of children from Burkina Faso compared to those from Italy. Not surprising, the dietary intakes were fundamentally different. Children in Burkina Faso consumed a predominantly plant-based vegetarian type diet, rich in fibers and plant-polysaccharides, low in fat, animal protein and simple sugars. On the other hand, children in Italy had a typical Western style diet. Similar patterns were seen comparing the adult gut microbiome of Westernized and traditional societies [140]. Although it is likely that dietary differences were the main factor leading to the distinct gut microbiomes, many other lifestyle, hygiene and ethnic differences between these populations may also have contributed to the observed differences. Among one-year-old infants of different ethnic background living in Canada, a higher abundance of lactic acid bacteria in South Asians and a higher abundance of genera within the order Clostridiales in Caucasians were observed [141]. However, the magnitude of these differences was relatively small and a link to their dietary habits could not be established.

Several studies examined associations between dietary intakes and gut microbiome in infants and children of varying ages within the same cultural background. In two Danish cohorts, protein and fiber intakes, as well as ‘family foods’, were positively associated with gut microbiota alpha diversity at 9 months, while fat intake showed an inverse association [142]. In Australian infants aged 6–24 months, dairy intake was positively associated with the Firmicutes to Bacteroidetes ratio but was inversely associated with microbial species richness and diversity [143]. In a recent study of Indian slum-dwelling infants aged between 10 and 18 months, a positive association of fat and iron intake on alpha diversity was observed [144]. In addition, a log ratio of *Lactococcus* to *Anaerococcus* was significantly higher in the group who reported consuming oils and fats. In children aged 1 to 2 years, Lachnospiraceae and Bacteroides were positively associated with fruit consumption, but inversely associated with processed meat and savory snacks [145]. In older children, aged 4–8 years, the consumption of food grouped into different dietary patterns were reported to have different microbiome profiles [146]. Specifically, the dietary patterns characterized by the intake of refined carbohydrates, and the intake of sugar and sweets showed an inverse association with gut microbiota alpha diversity indexes. On the other hand, the intake of fruit was positively correlated with the relative abundance of Bacteroides. In US children aged 2–9 years, the total fruit and fiber intake was associated with the relative abundance of *Lachnospira* species [147]. In another study of 5-year-old children, the strongest association observed was between the intake of nuts, seeds and legumes as well as meat and poultry and the microbiota community type characterized by a high abundance of *F. prausnitzii* [148]. *F. prausnitzii* is one of the age discriminant taxa increasing in abundance with the introduction of complementary diet, and it represents one of the most abundant anaerobic bacteria in the adult’s gut microbiota [149]. It is a main butyrate producer and plays an important role to maintain gut health [150]. Several nutrients have been reported to be associated with its abundance, among them the prebiotic fructooligosaccharides (FOS) [151,152]. As a strictly anaerobic species, *F. prausnitzii* copes with oxygen near the gut epithelia using a flavin-thiol electron shuttle [153]. Flavins, typical plant metabolites, may, therefore, favor the establishment of bacteria such as *Faecalibacterium*. Of note, low food diversity intake up to 1 year relates to a higher risk for asthma and food allergy later in life, which was also related to lower butyrate and propionate formation by the colonic gut microbiota [154,155]. Overall, only a few consistent patterns were observed, with a possible exception of the association of fruit intake with the abundance of Bacteroides, which was reported by two independent studies. However, most studies are small, and the dietary intakes are relatively homogenous among study participants making it difficult to find associations.

Several intervention studies with older infants and children tested supplementation with prebiotic fibers such as GOS and FOS, sometimes combined with probiotics [145]. Similar to observations made in younger infants, supplementing with prebiotic oligosaccharides resulted in a higher abundance of bifidobacteria [156,157,158]. Despite the modest effect size, the effect appeared to be quite consistent among the studies.

Infants and children with a dysbiotic gut microbiome, for example, upon antibiotic treatment, show a more pronounced response to microbiome active dietary supplementation [159]. In a randomized controlled trial conducted among 9- to 18-month-old children who were at risk for iron deficiency, the addition of vitamin E to an iron fortification was tested. Vitamin E was supposed to help reduce gut inflammation linked to an excess of iron intake. It is noteworthy that the vitamin E addition resulted in an increase in the relative abundance of *Roseburia* species, a typical butyrate producer [160]. As mentioned above mal- or undernourished children show an altered gut microbiome maturation. A series of elegant experiments combining results from observational studies with children and animal models led to the design of several microbiota-directed therapeutic foods. These were subsequently tested in children with severe acute malnutrition in Bangladesh. The combination of chickpea flour and raw banana, together with peanut flour and soy flour was the most effective in modulating the abundances of targeted bacteria for age-appropriate microbiome maturation, and additionally positively associated with biomarkers for growth [100,161].

Although the data are fragmented, nutritional guidance and designed nutritional solutions are a promising means to positively modulate the gut microbiome maturation trajectory of infants and children.

## 9. Outlook

The importance of an age-appropriate gut microbiome development and maturation is generally acknowledged. The extent to which an accelerated or delayed microbiome maturation profile affects the health of a child starts to be understood and individual microbial taxa and their metabolites emerge as causal agents. Despite the great progress in describing and understanding the early life gut microbiome, several key questions warrant to be understood. To this end, simultaneous investigation of the microbiome maturation trajectory in conjunction with immune and host–microbiota co-metabolites during early life may help to better define the neonatal windows of opportunity in humans.

The immune system development starts prenatally, and microbial metabolites and components from the mother are involved. What are the key factors and how do nutrition, environment and inherent physiological and genetic conditions affect these? Eventually, how malleable are prenatally set conditions after birth and how do the set conditions contribute to risks for health conditions later in life? During the exclusive breastfeeding period, a mother’s milk provides the reference standard for nutrition. Yet, C-section and antibiotics are relatively recent acquisitions of human culture. The question is, does a mother’s milk provide all that is needed to cope with those ‘new’ risks for an appropriate microbiome development? Reaching the age of needing complementary diets to fulfill an infant’s and child’s physiological needs, an inappropriate diet represents additional risks to an appropriate microbiome development. We have some understanding of what represents an appropriate microbiome in this age group, but many questions remain to be answered. How does the microbiome and co-developing immune system during the exclusive breastfeeding period drive subsequent development? Beyond fibers, what are other key nutrients, for which children often show inadequate intake, contributing to an appropriate microbiome for long-term health? Generally, for children starting a complementary diet, the knowledge on the impact of specific foods and dietary patterns is fragmented and warrants intensified research.

The extent to which the identified microbes and metabolic pathways are indicators of suboptimal host gut–microbiome mutualism and are causally linked to health conditions remains to be more generally established. It bears the obvious question to what extent corrective actions will work. Culture and tradition are key drivers in food choices influencing the maturing microbiome [102]. Importantly, which nutrients and food groups favor an appropriate gut microbiome maturation and eventually enable the development of appropriate immune competence are of broad research and public health interest.

## Figures and Tables

**Figure 1 microorganisms-09-02110-f001:**
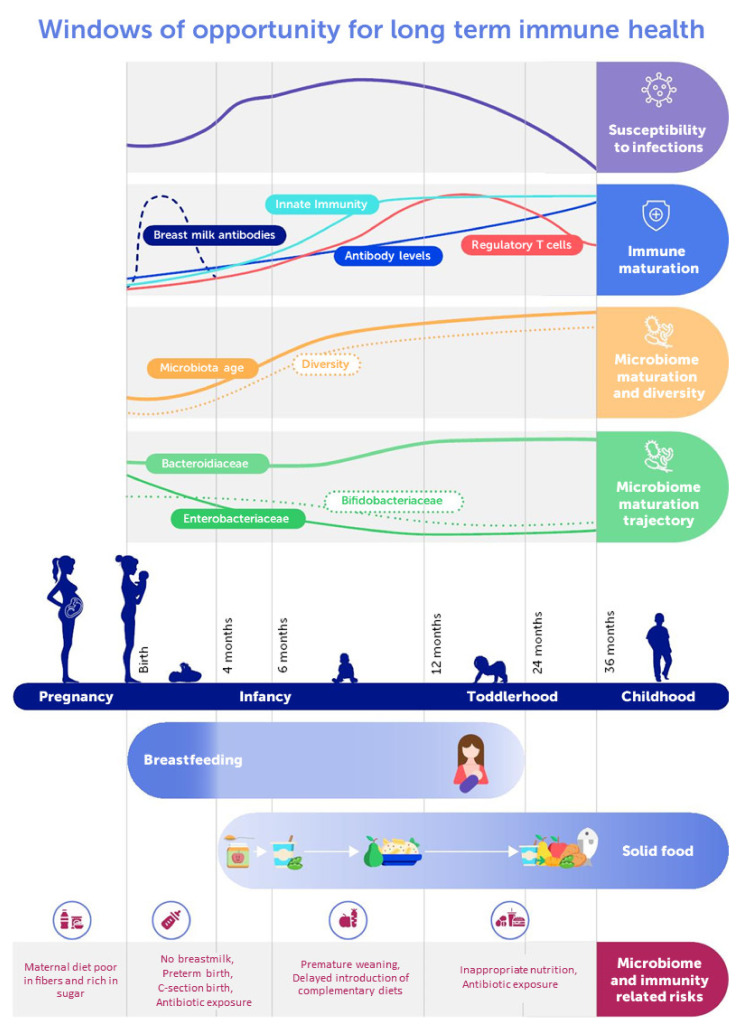
Schematic representation of the early life gut microbiome and immune development dynamics with feeding recommendations and major risk factors.

## Data Availability

Not applicable.

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
