# Peer review of "Nurturing the Early Life Gut Microbiome and Immune Maturation for Long Term Health"

_microorganisms, 2021, doi:10.3390/microorganisms9102110_

Round 1
Reviewer 1 Report
The narrative review "Nurturing the early life gut microbiome and immune matura- 2 tion for long term health" by Dogra et al., highlights some intriguing aspects related to newborns and the involvement of gut microbiota.
Since the main topic relies on the type of bacteria genera at each developmental step, I suggest including a table describing the taxa richness, the observed shifts, and increase/decrease trends.
Some observational studies like this one DOI: 10.3389/fcimb.2021.590202 may help in dealing with the central topic.
Author Response
Reviewer 1:
Comments and Suggestions for Authors
The narrative review "Nurturing the early life gut microbiome and immune maturation for long term health" by Dogra et al., highlights some intriguing aspects related to newborns and the involvement of gut microbiota.
Since the main topic relies on the type of bacteria genera at each developmental step, I suggest including a table describing the taxa richness, the observed shifts, and increase/decrease trends.
Some observational studies like this one DOI: 10.3389/fcimb.2021.590202 may help in dealing with the central topic.
Response: We thank the reviewer for this suggestion. We included an additional paragraph (Line 395) to highlight the taxa richness increase and changes over time for the mentioned key age discriminant taxa identified.
Reviewer 2 Report
Overall, this paper is a very complete body of work. It reads well, it is very thorough and timely as more and more questions and interests emerge within the field.
- Unclear sentence:
- Line 151. ‘The glycan exposed on mucous…. ’ – this sentence is unclear
- Odd words/grammar
- In line 54. ‘Learning to this end…’ as well as throughout the document the word ‘learnings’ is used a number of time, although correct it is not the conventional verbiage please consider ‘lessons’, ‘lessons learned’, or ‘understanding’
- Line 210-211. The term ‘below around 10 weeks’ and then in again ‘above around’, this is not correct. Please correct to ‘below 10 weeks’ or ‘below or equal to 10 weeks’, for as it stands it confuses the message
- In lines 160 – 161. Preterm born infants and term born infants are used, there is no need for the ‘born’ in either one
- Lines 163-165 and 219-221 are brilliant! Very important to include such details in the literature!!
- Please consider defining the term ‘immune silencing’.
- In your extensive discussion of the Bifidobacteria, when possible, could you please specify the species. As there appears to be a significant difference between the Bifidobacteria species on the context of infants.
- Please consider adding Henrik BM et al 2021 (https://doi.org/10.1016/j.cell.2021.05.030)
- Please consider what effect fetal microbiome would have on the developing immune and microbiome. There appears to be a debate around this topic with the a question of whether live microbe shape the fetal immune cell and the downstream effects which follow. https://www.cell.com/cell/fulltext/S0092-8674(21)00574-2.
Author Response
Reviewer 2:
Comments and Suggestions for Authors
Overall, this paper is a very complete body of work. It reads well, it is very thorough and timely as more and more questions and interests emerge within the field.
- Unclear sentence:
Line 151. ‘The glycan exposed on mucous…. ’ – this sentence is unclear
Response: We clarified the sentence and changed it as follows. “The glycans exposed on mucous feed specific microbes from inside,…” changed to “Specific microbes dwell on glycans exposed on mucous, which can be considered as feeding from inside, as opposed to feeding through diet, and this mechanism may greatly contribute to establishing an appropriate gut ecology.”
- Odd words/grammar
In line 54. ‘Learning to this end…’ as well as throughout the document the word ‘learnings’ is used a number of time, although correct it is not the conventional verbiage please consider ‘lessons’, ‘lessons learned’, or ‘understanding’
Line 210-211. The term ‘below around 10 weeks’ and then in again ‘above around’, this is not correct. Please correct to ‘below 10 weeks’ or ‘below or equal to 10 weeks’, for as it stands it confuses the message
Response: We changed the language as suggested line 54 “learnings” replaced with “lessons” and line 157 “learnings” replaced with “observations related to “. Line 210, “around” removed.
- In lines 160 – 161. Preterm born infants and term born infants are used, there is no need for the ‘born’ in either one
Response: We removed the word “born”.
- Lines 163-165 and 219-221 are brilliant! Very important to include such details in the literature!!
Response: We thank the reviewer for the feedback and we obviously fully agree.
- Please consider defining the term ‘immune silencing’.
Response: We thank the reviewer to bring our attention to a statement that may be misunderstood. We changed “immune silencing” with “immune homeostasis” to avoid misunderstanding with “immune suppression”.
- In your extensive discussion of the Bifidobacteria, when possible, could you please specify the species. As there appears to be a significant difference between the Bifidobacteria species on the context of infants.
Response: We included the species, when possible and provided in the references provided.
- Please consider adding Henrik BM et al 2021 (https://doi.org/10.1016/j.cell.2021.05.030)
Response: Henrik et al 2021 was already included (see former reference 39, now 41).
- Please consider what effect fetal microbiome would have on the developing immune and microbiome. There appears to be a debate around this topic with the a question of whether live microbe shape the fetal immune cell and the downstream effects which follow. https://www.cell.com/cell/fulltext/S0092-8674(21)00574-2.
Response: We thank the reviewer to mention this somewhat sensitive topic. We included a reflection on this highly debated topic (Line 91) with 2 additional references. This shifted all subsequent references as marked using track changes.